

# Epigenetic modification mechanism of histone demethylase KDM1A in regulating cardiomyocyte apoptosis after myocardial ischemia-reperfusion injury

Lin He, Yanbo Wang and Jin Luo

Department of Cardiology, The Center Hospital of Shaoyang, Shaoyang, China

## ABSTRACT

Hypoxia and reoxygenation (H/R) play a prevalent role in heart-related diseases. Histone demethylases are involved in myocardial injury. In this study, the mechanism of the lysine-specific histone demethylase 1A (KDM1A/LSD1) on cardiomyocyte apoptosis after myocardial ischemia-reperfusion injury (MIRI) was investigated. Firstly, HL-1 cells were treated with H/R to establish the MIRI models. The expressions of KDM1A and Sex Determining Region Y-Box Transcription Factor 9 (SOX9) in H/R-treated HL-1 cells were examined. The cell viability, markers of myocardial injury (LDH, AST, and CK-MB) and apoptosis (Bax and Bcl-2), and Caspase-3 and Caspase-9 protein activities were detected, respectively. We found that H/R treatment promoted cardiomyocyte apoptosis and downregulated KDM1A, and overexpressing KDM1A reduced apoptosis in H/R-treated cardiomyocytes. Subsequently, tri-methylation of lysine 4 on histone H3 (H3K4me3) level on the SOX9 promoter region was detected. We found that KDM1A repressed SOX9 transcription by reducing H3K4me3. Then, HL-1 cells were treated with CPI-455 and plasmid pcDNA3.1-SOX9 and had joint experiments with pcDNA3.1-KDM1A. We disclosed that upregulating H3K4me3 or overexpressing SOX9 reversed the inhibitory effect of overexpressing KDM1A on apoptosis of H/R-treated cardiomyocytes. In conclusion, KDM1A inhibited SOX9 transcription by reducing the H3K4me3 on the SOX9 promoter region and thus inhibited H/R-induced apoptosis of cardiomyocytes.

# INTRODUCTION

Ischemia-reperfusion (I/R) injury, first described in the 1980s, is a severe pathological process that occurs in multiple diseases, such as myocardial infarction, cerebral stroke, and surgical interventions, and is frequently associated with severe cellular damage and death (*Eltzschig & Eckle, 2011*; *Murry, Jennings & Reimer, 1986*; *Toldo et al., 2018*). Hypoxia refers to the intracellular oxygen content and pressure lower than normal levels, and would be triggered by factors such as hypoxemia, limited oxygen delivery, or impaired respiratory system (*MacIntyre, 2014*), while reoxygenation refers to the hypoxic cells regain oxygen and then the cells gradually recover to the normal state (*Chen et al.,*

Corresponding author
Lin He, helinnn1022@163.com

2020). Of note, the hypoxia–reoxygenation (H/R) process is known to cause neurological and cardiological damages and I/R injury (*Leon, Castillo & Gayubas, 2021*). On a separate note, myocardial ischemia-reperfusion injury (MIRI) after cardiovascular disorders (CVDs) accompanied by myocardial systolic dysfunction and cardiomyocyte necrosis and apoptosis would lead to fatal cardiovascular outcomes (*Lin et al., 2019*; *Gao et al., 2020*). Herein, we decided to probe the specific role of H/R in cardiomyocyte apoptosis after MIRI.

Histone methylation plays an important role in regulating gene expression and epigenetic inheritance of cells, serving as a part of the regulatory memory system of the epigenetic inheritance of cells (*Klose, Kallin & Zhang, 2006*). Furthermore, histone methylation is involved in myocardial injury induced by myocardial infarction (*Wang et al., 2018a*). However, the deletion of histone 3 lysine 4 methylation (H3K4) detriments the adaptive response to transverse aortic constriction and facilitate maladaptive cardiac remodeling (*Stein et al., 2015*). Additionally, the lysine-specific histone demethylase 1A (KDM1A/LSD1) is a demethylase first identified in 2004 that challenges the irreversibility of methylation signatures by demethylating H3K4 and H3K9 (*Ismail et al., 2018*; *Shi et al., 2004*). Prior studies demonstrated that KDM1A participates in cardiac development and diseases (*Davis et al., 2021*). KDM1A can increase the stability of hypoxia-inducible factor (HIF)-1α *via* demethylation of RACK1 and thus promotes cell adaptation to the hypoxic microenvironment (*Kim, Kim & Baek, 2021*; *Yang et al., 2017*). Moreover, LSD1 knockdown promotes apoptosis in multiple diseases (*Han et al., 2021*; *Zou et al., 2017*). Hence, we speculated that KDM1A may affect H/R-induced cardiomyocyte apoptosis through demethylating H3K4 methylation.

Sex Determining Region Y-Box Transcription Factors (SOXs) proteins are classified as a type of transcription factor possessing highly-conserved HMG box domain and could modulate developmental processes as well as homeostasis of adult tissues (*Zhang & Hou, 2021*). SOX9, belonging to the SOXs family, is a key transcription factor in the development and maturation of cartilage (*Lefebvre, Angelozzi & Haseeb, 2019*). A study has shown that microRNA-30e mitigates myocardial I/R injury and promotes ventricular remodeling through repressing SOX9 (*Cheng et al., 2021*). In addition, SOX9 has been shown to regulate many fibrosis-related genes and is activated in the case of ischemic injury (*Lacraz et al., 2017*). Our study aimed to verify the hypothesis we proposed that whether KDM1A can function on H/R-induced cardiomyocyte apoptosis by regulating SOX9 through demethylation of H3K4 methylation. Accordingly, in the follow-up experiments, the H/R cell models were established to induce cardiomyocyte apoptosis, and the expressions and roles of KDM1A and SOX9 in the models were detected.

## MATERIALS & METHODS

### Cell culture and H/R treatment

The mouse cardiomyocyte cell line HL-1 cells (GDC0606) were purchased from China Center for Typical Cultures Preservation (Wuhan, Hubei, China). HL-1 cells were cultured in RPMI-1064 medium (Gibco, Grand Island, NY, USA) supplemented with 10%

heat-inactivated fetal bovine serum (FBS, Gibco) at 37 °C with 5% $CO_2$. Subsequently, HL-1 cells were first cultured under hypoxia conditions (95% $N_2$ and 5% $CO_2$, 37 °C) for 2 h and then under oxygen conditions (95% air and 5% $CO_2$, 37 °C) for 12 h to establish the H/R cell models.

## Cell transfection

PcDNA3.1-KDM1A (oe-KDM1A; NM_001347221.1), pcDNA3.1-SOX9 (oe-SOX9; NM_011448.4) and the corresponding control empty plasmids pcDNA3.1-NC (oe-NC) were synthesized by Shanghai GenePharma Co., Ltd. (Shanghai, China). According to the manufacturer's instructions, 2 μg of pcDNA 3.1 plasmid was transfected into HL-1 cells using Lipofectamine 2000 (Invitrogen, Carlsbad, CA, USA) 48 h prior to H/R treatment. HL-1 cells were treated with CPI-455 (10 mM, a specific KDM5 inhibitor) to increase H3K4me3 level 30 min before H/R treatment. CPI-455 (Cat. No. S8287) was purchased from Selleck Chemicals (Selleck Chemicals, Houston, TX, USA).

## Enzyme-linked immunosorbent assay (ELISA)

HL-1 cells ($2 \times 10^5$ cells) were transfected and then used for establishing H/R models. Afterwards, the cell culture medium was centrifuged at 850 g for 3 min to collect the cell supernatant. The concentrations of lactate dehydrogenase (LDH), aspartate aminotransferase (AST), and creatine kinase-MB (CK-MB) in cell supernatant were measured using ELISA kits according to the manufacturer's instructions. LDH kits (YX-E20034) were purchased from Wuhan Yipu Biological Technology Co., Ltd (Wuhan, Hubei, China), CK-MB kit (KE1544) from ImmunoWay (Plano, TX, USA), and AST kit (ab263882) from Abcam (Cambridge, MA, USA).

## Caspase activity detection

The activity of Caspase-3 and Caspase-9 was determined using a colorimetric assay kit (Beyotime, Shanghai, China) according to the manufacturer's instructions. Briefly, HL-1 cells were collected and treated with 100 μL lysis buffer for 15 min. Then, the protein concentration was determined using a bicinchoninic acid (BCA) protein detection reagent (Beyotime, Shanghai, China). Subsequently, 150 μg of the lysates were incubated with 10 μL of Caspase-3 and Caspase-9 substrates (2 mM) at 37 °C for 4 h. The absorbance was measured at 405 nm using a microplate spectrophotometer (FLx800; BioTek, Winooski, Vermont, USA).

## Cell counting kit-8 (CCK-8) assay

The cell viability of HL-1 cells was detected using the CCK-8 assay kit (Beyotime, Shanghai, China) according to the manufacturer's instructions. HL-1 cells were loaded on 96-well plates at $2 \times 10^4$ cells/well and incubated for 24 h, 48 h, and 72 h, after which 10 μL CCK-8 reagent was added into each well for continuous incubation in the dark for 2 h. The absorbance of the reagent at a wavelength of 450 nm was calculated using a microplate reader (Hercules, CA, USA).

**Table 1  PCR primers.**

|  | Forward primer (5′–3′) | Reverse primer (5′–3′) |
|---|---|---|
| *KDM1A* | CGGCCCGAGATGTTATCT | AGGCCCCCGGGGGGCGAG |
| *SOX9* | GGCTCGCGTATGAATCTCCTG | TTCTTCAGATCCGGCTCG |
| Bax | GCGGCAGTGATGGACGGGTCC | AGCTCCATATTGCTGTCCAG |
| Bcl-2 | CGGGGAAGGATGGCGCAAGCC | GCAGCCATGTCCCGGTGC |
| GAPDH | AGAGGGATGCTGCCCTTACCC | AGTTGAGGTCAATGAAGG |

## Flow cytometry

Flow cytometry combined with the Annexin V/PI kit (BD Bioscience, San Jose, CA, USA) was used to detect HL-1 cell apoptosis. In short, HL-1 cells were collected, washed with PBS, and then re-suspended in 1 × binding buffer. Next, HL-1 cells were stained with Annexin V and PI in a dark room for 30 min. Flow cytometry (BD Bioscience) was performed through exciting lasers at 488 nm for flow cytometric analysis, and PI-stained HL-1 cells were collected using 575 nm optical filter configurations.

## Quantitative real-time polymerase chain reaction (qRT-PCR)

Total RNA was isolated from HL-1 cells using TRIzol (TaKaRa, Dalian, China) reagent and reversely transcribed into cDNA using PrimeScript RT kit (TaKaRa, Dalian, China). qRT-PCR amplification was performed using SYBR Green Real-time kit (TaKaRa, Dalian, China) on a CFX96 RT-PCR system (BioRad, Hemel Hempstead, Hertfordshire, UK) according to the producer's instructions. GAPDH was used as the endogenous control. The primers are shown in Table 1. The relative gene expression was calculated by the $2^{-\Delta\Delta Ct}$ method.

## Western blot

Total protein of histone was extracted using the histone extraction kit (#ab113476; Abcam). And, total protein of HL-1 cells was extracted using the cell lysis buffer RIPA and was quantified by BCA kits (Pierce, Rockford, IL, USA). The protein was isolated by sodium dodecyl sulfate-polyacrylamide gel electrophoresis (SDS-PAGE) and transferred to poly (vinylidene fluoride) (PVDF) membrane (Invitrogen, Carlsbad, CA, USA). The membrane was blocked with 5% skim milk and incubated with the primary antibodies anti-KDM1A (1:10,000; ab129195; Abcam), anti-H3K4me3 (1:1,000; ab213224; Abcam), anti-H3 (1: 1,000; ab1791; Abcam), and anti-GAPDH (1: 2,500; ab9485; Abcam) and secondary antibody (1:2,000; ab6728; Abcam). H3 was used as the internal reference for H3K4me3, and GAPDH was used as the internal reference for other proteins. The protein bands were observed using an enhanced chemiluminescence kit (Pierce).

## Chromatin immunoprecipitation (ChIP) assay

ChIP analysis was performed using EZ-ChIP (Millipore, Billerica, MA). Briefly, HL-1 cells were crosslinked in 1% formaldehyde for 10 min and lysed in SDS lysis buffer, and DNA was sheared by ultrasound using Covaris at 4 °C (5 min, 20 s on, 20 s off). After dilution of the lysates with ChIP buffer, immunoprecipitation was conducted with mouse IgG

(1:100; Ab6789, Abcam) and anti-H3K4me3 (1:20, ab213224, Abcam). The antibody-chromatin complex was precipitated with ChIP blocking protein G agarose at 4 °C for 1 h, followed by washing and elution. After reverse cross-linking of the protein-DNA complex, DNA was purified on a rotating column and analyzed by RT-PCR. SOX9 promoter PCR primers were as follows: forward 5′ AAGAGAGCATCATAAGGAGAC 3′ and reverse 3′ GGAGTATTTATTAGAGACCCT 5′.

## Statistical analysis

Statistical analysis of the data was performed using GraphPad Prism 8.0 statistical software (GraphPad Software Inc., San Diego, CA, USA). The measurement data were expressed in the form of mean ± standard deviation. The data between two groups were compared using the $t$ test. One-way ANOVA or two-way ANOVA was used for comparisons among multiple groups, and Tukey's multiple comparisons test was used for the post-hoc test. $p < 0.05$ indicated the difference was significant.

## RESULTS

### H/R treatment promotes cardiomyocyte apoptosis and downregulates KDM1A

In order to investigate the role of KDM1A in MIRI, we constructed *in vitro* MIRI cell models through treating HL-1 cells with H/R. After H/R treatment, the level of KDM1A was significantly reduced ($p < 0.05$, Figs. 1A–1B), HL-1 cell viability was reduced ($p < 0.05$, Fig. 1C), the concentrations of myocardial injury markers (LDH, AST, CK-MB) were increased ($p < 0.05$, Fig. 1D), the level of pro-apoptotic protein (Bax) was increased, the level of anti-apoptotic protein (Bcl-2) was decreased ($p < 0.05$, Fig. E), the apoptosis rate was notably increased ($p < 0.05$, Fig. 1F), and the activities of Caspase-3 and Caspase-9 proteins were significantly increased ($p < 0.05$, Fig. 1G). The above results indicated that H/R treatment promoted the apoptosis of cardiomyocytes treated by H/R and downregulated the expression of KDM1A.

### Overexpressing KDM1A reduces apoptosis of H/R-treated cardiomyocytes

To further study the role of KDM1A in H/R-induced cardiomyocyte apoptosis, HL-1 cells were transfected with pcDNA3.1-KDM1A (oe-KDM1A) ($p < 0.05$, Fig. 2A). In HL-1 cells overexpressing KDM1A, KDM1A protein level was elevated ($p < 0.05$, Fig. 2B), cell viability was increased ($p < 0.05$, Fig. 2C), the levels of LDH, AST, and CK-MB were decreased ($p < 0.05$, Fig. 2D), Bax mRNA expression was decreased, Bcl-2 mRNA expression was increased ($p < 0.05$, Fig. 2E), the apoptosis rate was decreased ($p < 0.05$, Fig. 2F), and the activities of Caspase-3 and Caspase-9 were significantly decreased ($p < 0.05$, Fig. 2G). The above results indicated that overexpression of KDM1A reduced apoptosis of H/R-treated cardiomyocytes.

### KDM1A inhibits the transcription of SOX9 by reducing H3K4me3

It has been reported that KDM1A could inhibit the expressions of its downstream genes by removing the H3K4me3 markers from the gene promoters (*Zhao et al., 2020*). In order

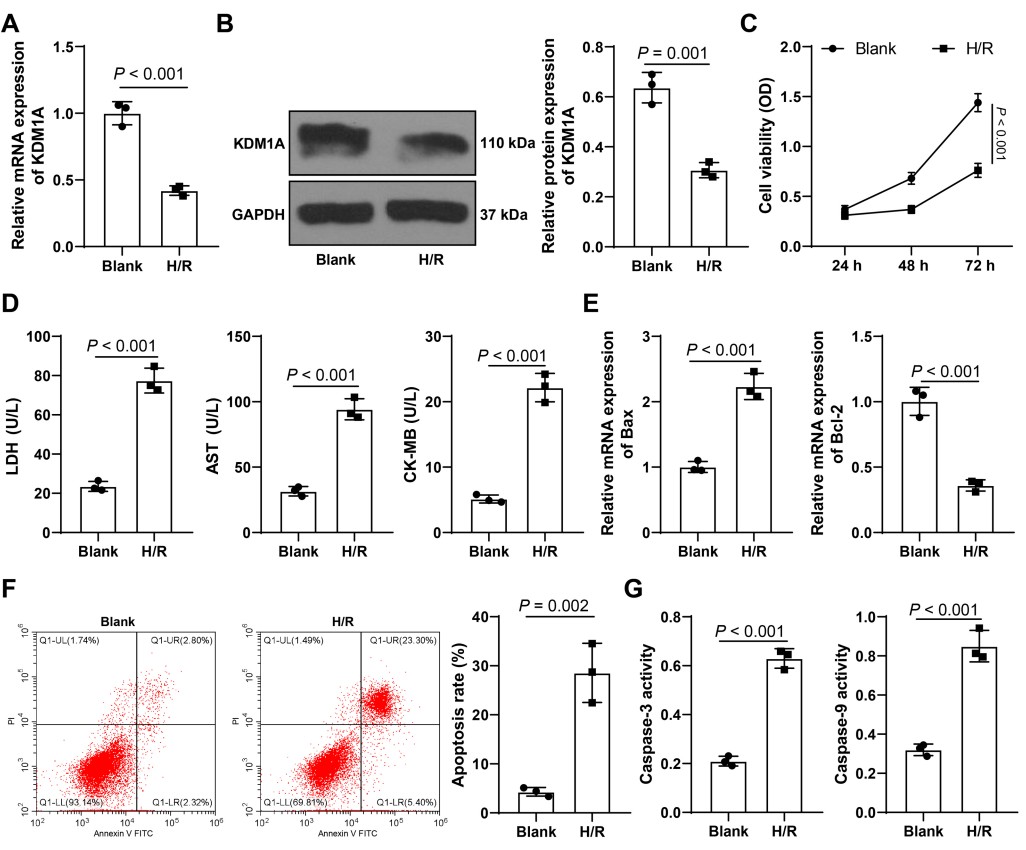

**Figure 1** **H/R treatment promotes cardiomyocyte apoptosis and downregulates KDM1A.** HL-1 cells were subjected to hypoxia-reoxygenation treatment (the H/R group), with HL-1 cells untreated as the control (the Blank group). (A–B) qRT-PCR and Western blot were adopted to detect the levels of KDM1A in HL-1 cells; (C) CCK-8 was used to verify HL-1 cell viability; (D) ELISA was performed to detect the concentrations of myocardial injury markers in HL-1 cells; (E) qRT-PCR was used to detect the mRNA levels of apoptotic proteins in HL-1 cells; (F) Flow cytometry was conducted to detect the apoptosis rate of HL-1 cells; (G) Colorimetric method was used to examine the activities of Caspase-3 and Caspase-9. The cell experiment was repeated three times independently; data were expressed as mean ± standard deviation. Data in (A–B) and (D–G) were tested by independent $t$ test; data in (C) were analyzed by two-way ANOVA, followed by Tukey's post-hoc test. LDH, Lactate Dehydrogenase; AST, Aspartate Aminotransferase; CK-MB, Creatine Kinase-MB.

to explore the downstream mechanism of KDM1A in MIRI, we performed Western blot to test the level of H3K4me3. The results showed that the level of H3K4me3 was increased in H/R-treated HL-1 cells and was decreased after overexpressing KDM1A ($p <$ 0.05, Fig. 3A). Additionally, H3K4me3 reduction can inhibit the transcription of SOX9 (*Wang et al., 2018b*), and SOX9 is upregulated in MIRI (*Cheng et al., 2021*). Therefore, we speculated that KDM1A in H/R cells could inhibit the transcription of SOX9 by reducing H3K4me3, thereby affecting apoptosis of cardiomyocytes after MIRI. The above speculation was subsequently supported by ChIP analysis, which showed that H3K4me3 was enriched on the SOX9 promoter but not on the SOX9 exon, and overexpressing KDM1A reduced the level of H3K4me3 on the SOX9 promoter ($p <$ 0.05, Fig. 3B). The

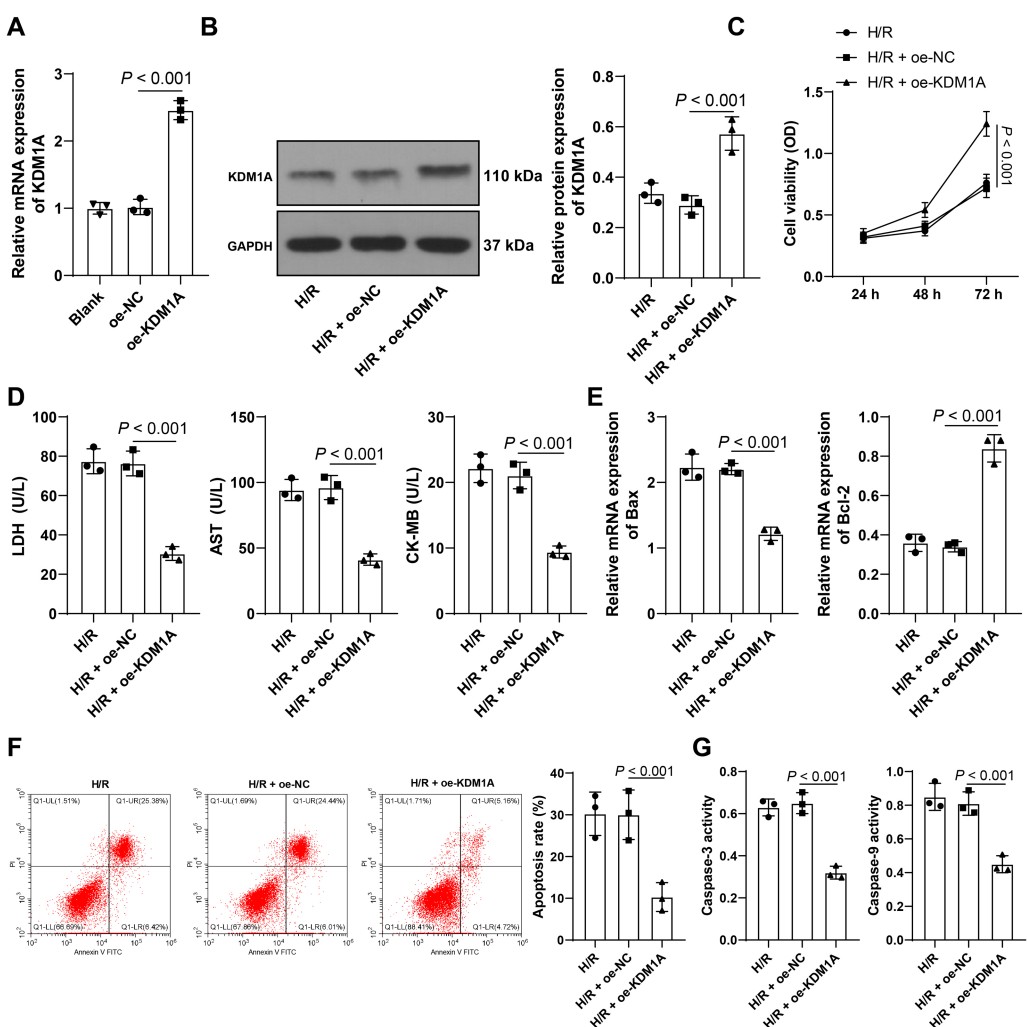

**Figure 2** **Overexpressing KDM1A reduces apoptosis of H/R-treated cardiomyocytes.** Plasmid pcDNA3.1-KDM1A (oe-KDM1A) was transfected into HL-1 cells, with pcDNA3.1-NC (oe-NC) as the control. (A) qRT-PCR was used to measure the transfection efficiency; (B) Western blot was performed to examine the protein level of KDM1A in HL-1 cells; (C) CCK-8 assay was adopt to detect cell viability; (D) ELISA was conducted to the concentrations of myocardial injury markers in HL-1 cells; (E) qRT-PCR was performed to verify the mRNA levels of apoptotic proteins in HL-1 cells; (F) Flow cytometry was used to detect the apoptosis rate; (G) Colorimetric method was conducted to measure the activities of Caspase-3 and Caspase-9. The cell experiment was repeated three times independently; data were expressed as mean ± standard deviation. Data in (C) were analyzed using two-way ANOVA and data in (A–B) and (D–G) were analyzed using one-way ANOVA, followed by Tukey's post-hoc test. LDH, Lactate Dehydrogenase; AST, Aspartate Aminotransferase; CK-MB, Creatine Kinase-MB.

level of SOX9 mRNA was increased in H/R-treated HL-1 cells and was decreased after overexpressing KDM1A ($p < 0.05$, Fig. 3C). The above results indicated that KDM1A could inhibit the transcription of SOX9 by reducing H3Kme3 on the SOX9 promoter.

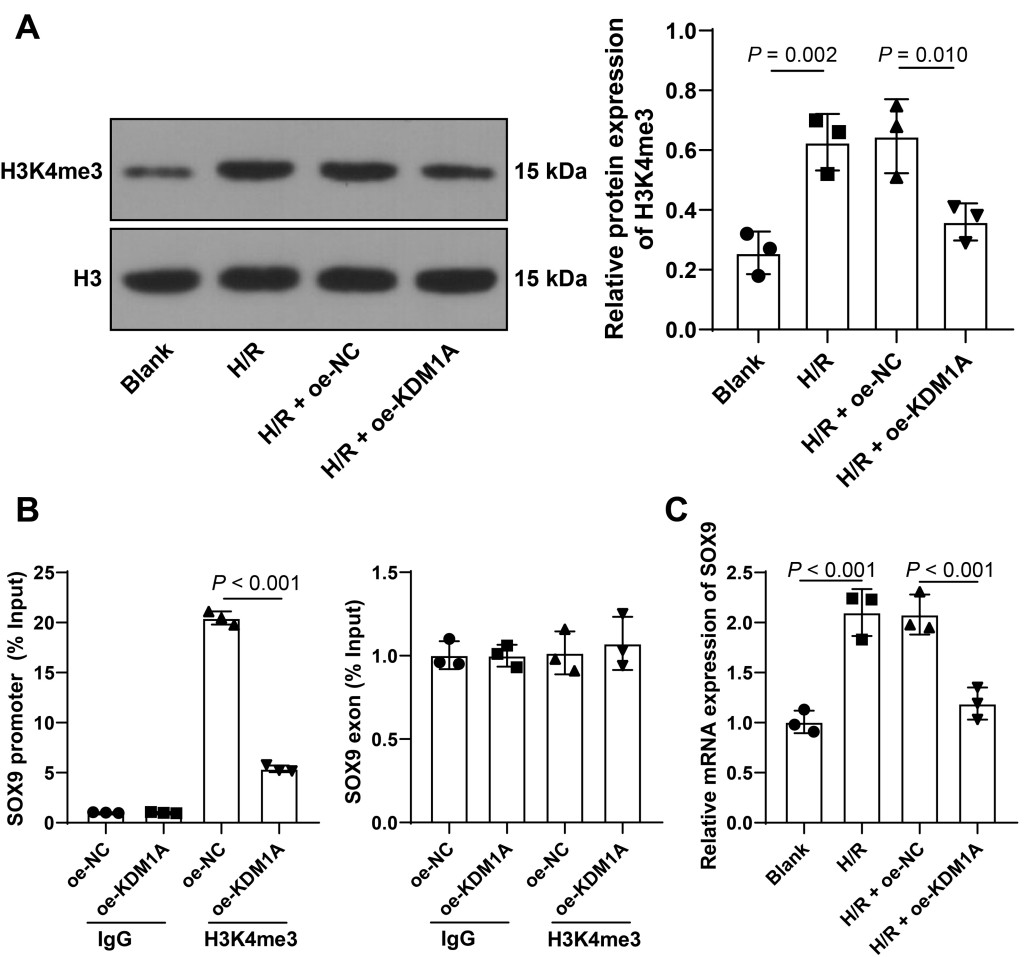

**Figure 3** **KDM1A inhibits the transcription of SOX9 by reducing H3K4me3.** (A) Western blot was used to detect the level of H3K4me3 in HL-1 cells. (B) ChIP analysis was used to detect the changes of H3K4me3 on the SOX9 promoter and exon; (C) qRT-PCR to detect the level of SOX9 mRNA in HL-1 cells; The cell experiment was repeated three times independently; data were expressed as mean ± standard deviation. Data in (B) were analyzed by two-way ANOVA and data in (A) and (C) were analyzed by one-way ANOVA, and followed by Tukey's post-hoc test.

## Upregulating H3K4me3 reverses the effect of overexpressing KDM1A on alleviating H/R-induced cardiomyocyte apoptosis

To prove that KDM1A limited apoptosis of H/R-treated HL-1 cells by reducing H3K4me3, we used CPI-455 to promote H3K4me3 in HL-1 cells transfected with oe-KDM1A. The results showed that, compared with HL-1 cells overexpressing KDM1A alone, in the H/R + oe-KDM1A + CPI group, the level of H3K4me3 was increased ($p < 0.05$, Fig. 4A), the mRNA expression of SOX9 was elevated ($p < 0.05$, Fig. 4B), HL-1 cell viability was decreased ($p < 0.05$, Fig. 4C), the levels of LDH, AST, and CK-MB were increased ($p < 0.05$, Fig. 4D), Bax mRNA level was increased, Bcl-2 mRNA level was decreased ($p < 0.05$, Fig. 4E), the apoptosis rate was increased ($p < 0.05$, Fig. 4F), and the activities of Caspase-3 and Caspase-9 were increased ($p < 0.05$, Fig. 4G). Briefly, upregulating

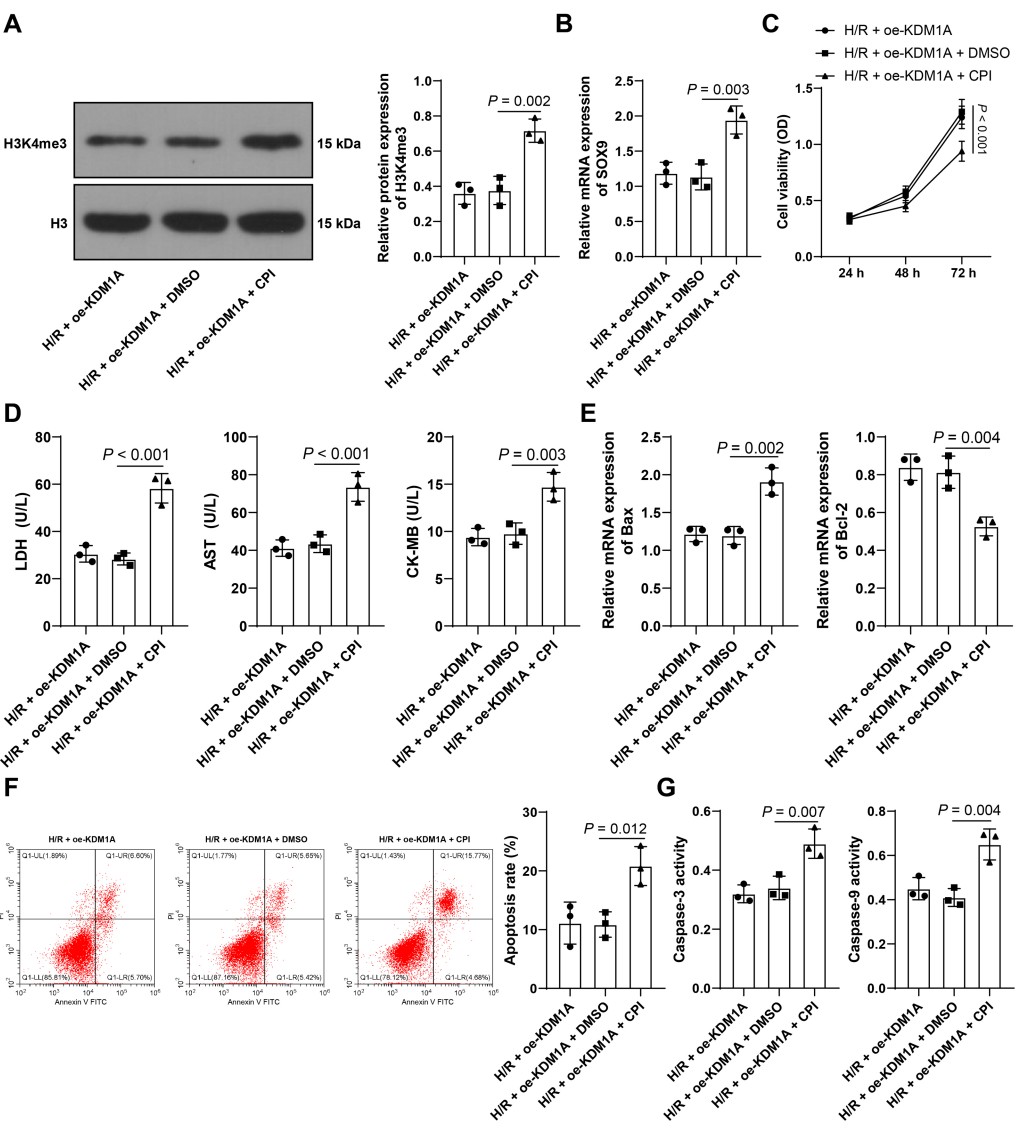

**Figure 4** **Upregulating H3K4me3 reverses the effect of overexpressing KDM1A on alleviating H/R-induced cardiomyocyte apoptosis.** H3K4me3 in H/R cells was promoted using CPI-455 (solvent control: DMSO) and then the cells were combined with oe-KDM1A. (A) Western blot was used to detect the levels of H3K4me3 in HL-1 cells; (B) qRT-PCR was performed to examine the mRNA expression of SOX9 in HL-1 cells; (C) CCK-8 was adopted to detect cell viability; (D) ELISA was performed to verify the concentrations of myocardial injury markers in HL-1 cells; (E) qRT-PCR was conducted to examine the mRNA levels of apoptotic proteins in HL-1 cells; (F) Flow cytometry was used to detect the apoptotic rate; (G) Colorimetric method was to conducted to test the activities of Caspase-3 and Caspase-9 in the cells. The cell experiment was repeated three times independently; data were expressed as mean ± standard deviation. Data in (C) were analyzed by two-way ANOVA and data in (A–B) and (D–G) were analyzed by one-way ANOVA, and followed by Tukey's post-hoc test. LDH, Lactate Dehydrogenase; AST, Aspartate Aminotransferase; CK-MB, Creatine Kinase-MB.

H3K4me3 could reverse the inhibition of overexpression of KDM1A on H/R-induced cardiomyocyte apoptosis.

## Overexpressing SOX9 reverses the alleviative effect of overexpressing KDM1A on H/R-induced cardiomyocyte apoptosis

To prove that SOX9 may be involved in the regulation of KDM1A on H/R-induced HL-1 cell apoptosis, pcDNA3.1-SOX9 (oe-SOX9) was transfected into HL-1 cells and the transfection efficiency was verified ($p < 0.05$, Fig. 5A). And, the transfected HL-1 cells underwent a joint experiment with oe-KDM1A. The results showed that compared with the group transfected with oe-KDM1A alone, in the H/R + oe-KDM1A + SOX9 group, HL-1 cell viability was decreased ($p < 0.05$, Fig. 5B), the concentrations of LDH, AST, and CK-MB were increased ($p < 0.05$, Fig. 5C), Bax mRNA level was increased, Bcl-2 mRNA level was decreased ($p < 0.05$, Fig. 5D), the apoptosis rate was increased ($p < 0.05$, Fig. 5E), and Caspase-3 and Caspase-9 activities were increased ($p < 0.05$, Fig. 5E). The above results indicated that overexpressing SOX9 could reverse the effect of overexpression of KDM1A on apoptosis of H/R-treated HL-1 cells.

## DISCUSSION

MIRI is regarded as the primary manifestation of CVDs (*Hausenloy & Yellon, 2013*). H/R process is considered a pivotal trigger factor for CVDs and exerts an irreversible impairment to homeostasis, causing cardiomyocyte apoptosis and injury, heart failure, or other detrimental outcomes (*Chen et al., 2021*). In addition, histone demethylases KDMs are proven to modulate the cellular processes of cardiomyocytes and be implicated with CVD progression (*Akerberg et al., 2017*; *Mokou et al., 2019*). Herein, our investigation in this work highlighted the regulatory functions of KDM1A on cardiomyocyte apoptosis after MIRI.

It has been reported that H/R induces cardiomyocyte apoptosis and inflammation (*Chen et al., 2021*; *Fornes et al., 2020*). Notably, a previous study illustrated that changes in KDM1A expression in neonatal mice could affect cardiomyocyte proliferation as well as myocardial regeneration through demethylation (*Fei et al., 2021*). In our study, HL-1 cells were subjected to H/R to establish MIRI cell models. Afterwards, the expression of KDM1A in H/R-induced HL-1 cells was decreased. Besides, HL-1 cell viability was decreased, LDH, AST, and CK-MB concentrations were increased, and HL-1 cell apoptosis was promoted. Through literature review, the decreased expressions of cardiomyocyte injury markers, LDH, AST, and CK-MB indicates the improvement of injury (*Chen et al., 2018*; *Yu et al., 2020*). And, many previous studies have proven the promotion of H/R on cardiomyocyte apoptosis and injury (*Lu, Bu & Yun, 2019*; *Wang et al., 2021*). The above indicated that H/R treatment promoted apoptosis of H/R-treated cardiomyocytes and downregulated KDM1A. Subsequently, we overexpressed KDM1A in HL-1 cells, after which, HL-1 cell viability was increased, myocardial injury was mitigated, and apoptosis was decreased. A relevant study has shown that KDM1A upregulation permits autophagy within the appropriate range, thereby reducing apoptosis of H/R-treated cardiomyocytes

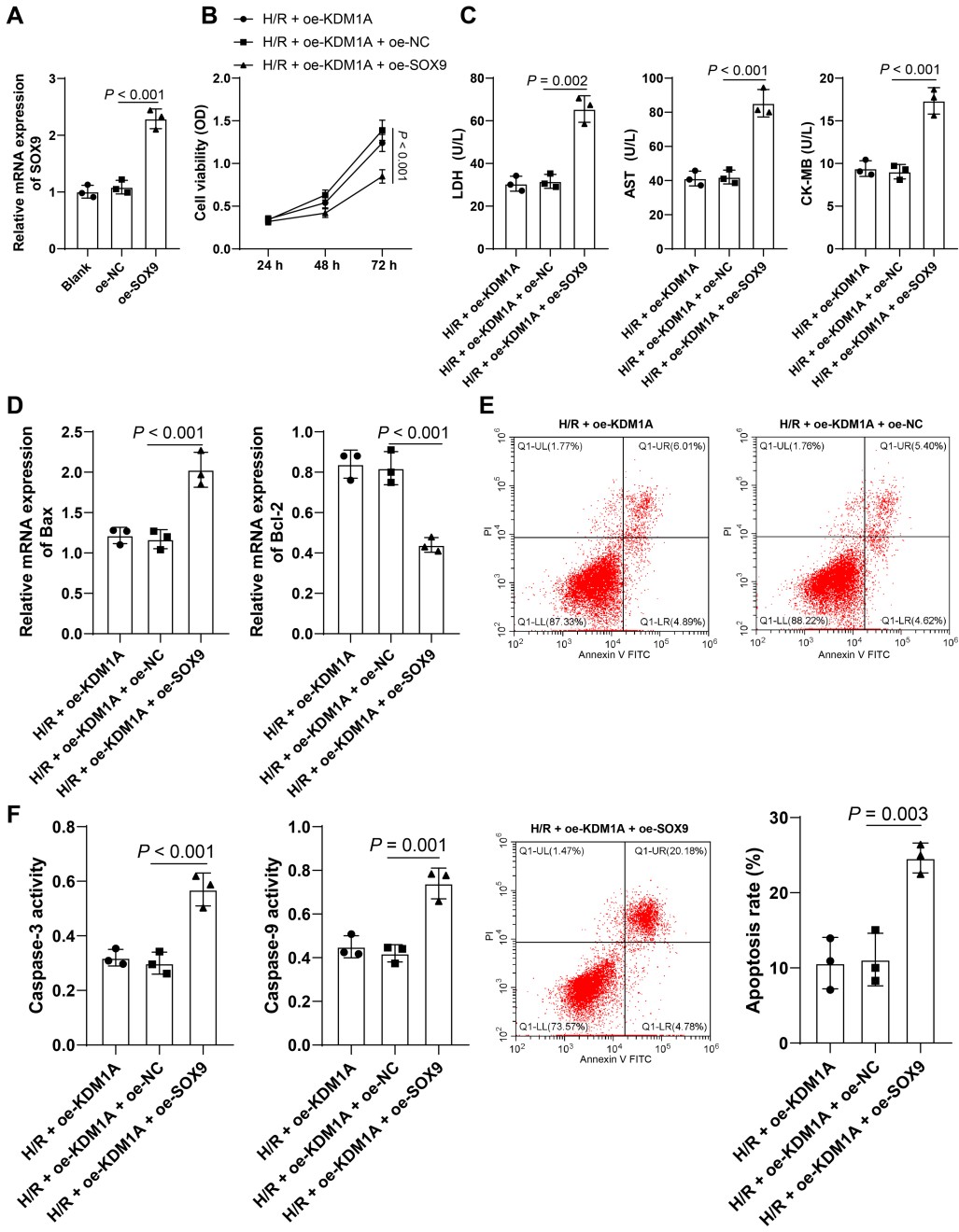

**Figure 5  Overexpressing SOX9 reverses the alleviative effect of overexpressing KDM1A on H/R-induced cardiomyocyte apoptosis.** pcDNA3.1-SOX9 (oe-SOX9) was transfected into HL-1 cells using oe-NC as the control. (A) qRT-PCR was adopted to detect the transfection efficiency. Then, the H/R-treated HL-1 cells overexpressing SOX9 were combined with oe-KDM1A. (B) CCK-8 was used to detect cell viability; (C) ELISA was used to detect the concentrations of myocardial injury markers in HL-1 cells; (D) qRT-PCR was conducted to verify the mRNA levels of apoptotic proteins in HL-1 cells; (E) Flow cytometry was used to examine the apoptotic rate; (F) Colorimetric method was adopted to test that activities of Caspase-3 and Caspase-9 proteins in HL-1 cell. The cell experiment was repeated 3 times independently; data were expressed as mean ± standard deviation. Data in (B) were analyzed by two-way ANOVA and data in (A) and (C–F) were analyzed by one-way ANOVA, followed by Tukey's post-hoc test. LDH, Lactate Dehydrogenase; AST, Aspartate Aminotransferase; CK-MB, Creatine Kinase-MB.

(*Song et al., 2018*), which supported our results that overexpression of KDM1A reduced apoptosis of H/R cardiomyocytes.

H3K4me3 is closely related to several heart-related pathologies, such as hypertrophic/dilated cardiomyopathy (*Raveendran et al., 2020*; *Tran et al., 2022*), and is associated with the conversion of the induced cardiomyocytes (*Liu et al., 2016*). Additionally, dexmedetomidine precondition in lung tissues alleviated I/R-induced lung injury *via* modulating H3K4me3 modification on the KGF-2 promoter (*Hong et al., 2021*). KDM1A may function on MIRI through demethylating H3K4me3 in MIRI. Accordingly, we found that the level of H3K4me3 was increased in H/R-treated HL-1 cells and was decreased after overexpressing KDM1A. On top of that, a prior study stated that KDM6A knockdown elevates H3K27me3 level but decreases H3K4me3 level on SOX9 promoter (*Wang et al., 2018b*) and SOX9 inhibition could reduce apoptosis of hypoxia-treated cardiomyocytes (*Rui et al., 2022*). Hence, we speculated that KDM1A might inhibit the transcription of SOX9 and thus affect apoptosis of H/R-treated cardiomyocytes by reducing H3K4me3. As expected, overexpressing KDM1A reduced the level of H3K4me3 on the SOX9 promoter, and decreased the mRNA level of SOX9. Besides, a previous study has found that KDM1A bound to Hotair could inhibit the molecule transcription by reducing the H3K4me3 level located on the Hotair promoter (*Zhao et al., 2020*). Overall, the above results indicated that KDM1A could inhibit the transcription of SOX9 by reducing H3K4me3 on the SOX9 promoter region.

CPI-455 could be used as a KDMs inhibitor, and CPI-455-mediated KDM inhibition elevates the global levels of H3K4me3 (*Liu et al., 2019*; *Vinogradova et al., 2016*). Thereafter, CPI-455 was used to promote H3K4me3 in H/R-treated HL-1 cells overexpressing KDM1A. After H3K4me3 promotion, HL-1 cell viability was significantly reduced, myocardial injury was aggravated, and cell apoptosis was increased. It has been verified that H3K4me3 inhibition could alleviate I/R-induced renal and limit fibrosis and inflammation (*Shimoda et al., 2019*). The above result demonstrated that H3K4me3 upregulation could reverse the effect of overexpressing KDM1A on cardiomyocyte apoptosis after MIRI.

In addition, previous studies have proven that H/R process could modulate SOX9 expression in rat chondrocytes and the interaction between KDM1A and SOX9 in endochondral ossification (*Li et al., 2018*; *Sun et al., 2020*). Subsequently, HL-1 cells were transfected with oe-SOX9 and received a joint experiment with oe-KDM1A to probe the involvement of SOX9 in the regulation of KDM1A on H/R-treated cardiomyocyte apoptosis. Afterwards, we found that overexpressing SOX9 reduced HL-1 cell viability, promoted myocardial injury, and facilitated cell apoptosis. Previous literature showed that SOX9 suppression encourages cell activity and limits inflammation apoptosis in H/R-treated cardiomyocytes and further alleviated MIRI (*Cheng et al., 2021*), while overexpression of SOX9 limits cell activity and thus promotes hepatic I/R (*Fan et al., 2018*). These findings made it plausible that overexpressing SOX9 could reverse the mitigating effect of overexpressing KDM1A on cardiomyocyte apoptosis in MIRI.

## CONCLUSIONS

In conclusion, we revealed that KDM1A inhibited the transcription of SOX9 by reducing H3K4me3 on the SOX9 promoter region, and thereby inhibited H/R-induced cardiomyocyte apoptosis. We hope our study can provide a promising direction for the research and treatment related to MIRI. However, our study still has its limitations. We only investigated the demethylation of KDM1A on H3K4me3 located on the SOX9 promoter region and failed to study other histone demethylases. We verified the KDM1A/H3K4me3/SOX9 mechanism only in the *in vitro* models established by a single cell type. Going forward, we will conduct further studies to probe other sites of KDM1A for histone demethylation, to explore the roles of other histone demethylases in cardiomyocyte apoptosis after MIRI. Accordingly, we will use different types of cardiomyocytes, such as H9C2 cells, to establish MIRI cell models and will validate our study in the *in vivo* animal models.

### Funding
The authors received no funding for this work.

### Competing Interests
The authors declare there are no competing interests.

### Author Contributions
- Lin He conceived and designed the experiments, performed the experiments, authored or reviewed drafts of the article, and approved the final draft.
- Yanbo Wang performed the experiments, analyzed the data, prepared figures and/or tables, and approved the final draft.
- Jin Luo performed the experiments, analyzed the data, authored or reviewed drafts of the article, and approved the final draft.

### Data Availability
The raw data are available in the Supplemental Files.

### Supplemental Information
Supplemental information for this article can be found online at http://dx.doi.org/10.7717/peerj.13823#supplemental-information.

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
