# Peer review of "Epigenetic modification mechanism of histone demethylase KDM1A in regulating cardiomyocyte apoptosis after myocardial ischemia-reperfusion injury"

_PeerJ, doi:10.7717/peerj.13823_

## Round 0.1 · original submission · Major Revisions

You will see that the three Reviewers agreed that your manuscript is of the caliber that merits consideration by PeerJ. However, they identified key areas that need revision to further improve the manuscript. I urge you to revise the manuscript, paying close attention to the suggestions of the Reviewers.

Reviewer 1 ·

Basic reporting

In this work, they revealed the mechanism by which KDM1A inhibits SOX9 transcription by reducing H3K4me3 in the SOX9 promoter region and thus promotes cardiomyocyte apoptosis after MIRI. By and large, the language of the manuscript is of good quality and the pictures are well laid out.

Experimental design

1.In the abstract, “HE-1 cells” or “HL-1 cells”?
2.In cell transfection, they didn’t indicate the time of cell transfection. How long were the cells transfected before CIP-455 was used and how long was the CIP-455 treatment? What’s the rationale for using CIP-455 to treat cells?
3.In ELISA assay, they should describe how the cell supernatant was obtained.
4.In ine 128, they should detail the relationship between the time of transfection and the time of model processing.
5.What treatments were the cells subjected to before experiments such as ELISA were performed? The relevant information is severely lacking in the manuscript.
6.In line 215, after the transfection of pcDNA3.1-KDM1A, the protein expression of KDM1A needs to be detected.
7.More details about Blank are needed. And what is “CPI”?
8.In line 241, changes of SOX9 should be detected after CPI-455 treatment.
9.Since the relationship between KDM1A and SOX9 has been reported (line 321), I wonder what is the innovative point of this study then?
10.They performed their experiments in cell models, however, they didn’t design relevant animal experiments to verify their results.

Validity of the findings

no comment

Additional comments

no comment

·

Basic reporting

In general, the overall quality of the manuscript is impressive and the study content meets the requirements of Peer J. The introduction and background sections provide the necessary information and the picture quality is up to standard. Besides, description of the method and the results in the manuscript is clearly presented and shows enough details for other researchers to refer. However, there are some details that need to be revised.

Experimental design

1.Line 44. What is “HE-1 cells”? The cells included in this study may be HL-1 cells, I guess.
2.Line 45. Please give the full name of “SOX9”.
3.Line 128. What is “pcDNA3.1-NC”?
4.Line 135. The number of cells used in ELISA assay should be given.

Validity of the findings

5.line 335-342. The authors should further discuss the limitations of the study in the conclusion part; after all, this study is based on cell experiments and the authors didn’t validate this mechanism in any animal or clinical experiment.
6.Fig 1-5. Please give the full name of abbreviations in graphs in the figure notes (such as LDH, AST, etc.). And, since the specific p-value is shown in Figs, "* p < 0.05" should be deleted from the figure notes.
7.Fig.2. After overexpressing KDM1A, Western blotting for detecting its protein expression is needed.
8.Fig.5. The P-value in 5F is displayed incorrectly, and the strikethrough is misused as an underscore.

Reviewer 3 ·

Basic reporting

The authors wrote a short yet complete manuscript. While this is overall comprehensible in all its parts, I must recommend English language editing to improve the clarify of certain sections, particularly the abstract, introduction, and conclusions, which can be hard to parse. There are also several typos (i.e. Line 91: two extra characters at the start; Line 99: missing space after H3K4). Of note, I encourage the authors to be more specific when describing earlier findings (i.e. Lines 94 and 96: clarify what "implicate" and "is necessary' mean in this context by indicating the specific findings of the references cited therein; Line 110: what does "regulate" mean here? Promote/inhibit?)

The literature cited is sufficient yet I notice a bias towards very recent reports. I recommend that more attention is based to the seminal findings in the field: for instance, I/R injury was first described in the 80's (PMID: 3769170).

The figures are well structured, but the authors should modify all bar graphs to show overlayed individual data points as per the policy of PeerJ. I also could not find the raw data for Figures 4 and 5 in the same way done for Figs. 1-3; please amend to allow their examination. Furthermore, WB raw data are provided only for the representative WB experiment; please supply the images also for the other replicates used for quantifications.

Experimental design

The question being studied is clearly stated and of interest, though the ambition of the study is very limited in looking at the regulation of a specific gene rather than the global changes that are induced following KDM5A down regulation in H/R. The authors do admit this in the conclusions, though.

From an experimental perspective my main concern is that atrial-like HL-1 cells are not a great model for H/R: not only they are hardly comparable to normal cardiomyocytes, but they are less sensitive to H/R than other more physiologically-relevant cardiac cell lines such as ventricular-like H9C2, see for reference PMID: 25450968. Authors are encouraged to validate key findings in this second cell line or, even better, in iPSC-derived cardiomyocytes and/or primary cardiomyocytes

Other important remarks:

H3K4me3 levels by WB should be presented alongside total H3, which is a better loading control than GAPDH. Indeed, consistent extraction of histones is challenging using conventional buffers such as RIPA that have physiologic pH: acid extraction is the preferred method for accurate histone WB, see PMIDs: 17545981 and this protocol from Abcam, the supplier the authors use for their histone antibodies: https://www.abcam.com/protocols/histone-extraction-protocol-for-western-blot

I am confused as to why in Fig 2E the gating thresholds for the oe-KDM1A condition are very different from the control and NC conditions: were these experiments run on separate days? If not what else explains the overall increased Annexin V and PI signals in the KDM1A conditions? A similar inconsistency is noted in Fig 4E.

I noticed Figs 1 and 2 use the vewry same raw data for the Blank and H/R conditions: were the experiments run at the same time?

The raw data for what the authors define as biological replicates are exceedingly close to one another, much more than it can be possible in my experience. This is particularly true for RT-qPCR Ct values, which are all within 0,5-1 of one another for both the housekeeping and genes of interest. Perhaps the authors consider biological replicates as multiple wells from the same culture? If so I recommend that key experiments are repeated at least twice in separate cultures (i.e. different passages of HL1 cells), so as to account for inter-experimental variability rather than intra-experimental one, which is much less meaningful.

In Fig. 3B findings would be strengthened by also reporting results for an unrelated genomic region (i.e. a non H3K4me3-marked domain on or around SOX9).

Finally, I have several requests for clarifications regarding the methods:
- Line 128: please provide the RefSeq ID for the specific cDNAs (and if possible deposit the plasmids to Addgene as per PeerJ policies).
- Line 128: I assume that pcDNA3.1-NC is for negative control - was this an empty plasmid? If so using an unrelated protein would have been preferable as control and could be added for at least key experiments
- Line 163: please provide further details about the cytometer used, for instance the lasers and optical filter configurations
- Line 182: for the loading control did the author strip and reprobe the same membrane used the protein of interest? Or were membranes cut in pieces based on the anticipated MW?
- Line 187: clarify sonicator model and conditions used
- Table 1: the size of some RT-qPCR amplicons seems way too big (i.e. 2.6 kb for KDM1A). Please check the sequences or clarify

Validity of the findings

As mentioned above I could not examine the raw data for Figs 4-5 as well as raw data for all WB. I also await for clarification on experimental and analytical designs that are key to make a complete assessment of the robustness of the study.

From the figures presented the main conclusion of the study appears to be generally supported, and there is no ove-rinterpetation of the data.

Additional comments

NA

---

## Round 0.2 · accepted · Accept

We look forward to your next submission.

Reviewer 1 ·

Basic reporting

no comment

Experimental design

no comment

Validity of the findings

no comment

·

Basic reporting

No comment.

Experimental design

No comment.

Validity of the findings

No comment.

Additional comments

No comment.